# Beyond Ramen: Investigating Methods to Improve Food Agency among College Students

**DOI:** 10.3390/nu13051674

**Published:** 2021-05-14

**Authors:** Lizzy Pope, Mattie Alpaugh, Amy Trubek, Joan Skelly, Jean Harvey

**Affiliations:** 1Department of Nutrition and Food Sciences, University of Vermont, Burlington, VT 05405, USA; efpope@uvm.edu (L.P.); Amy.Trubek@uvm.edu (A.T.); Jean.Harvey@uvm.edu (J.H.); 2Department of Medical Biostatistics, University of Vermont, Burlington, VT 05405, USA; Joan.Skelly@uvm.edu

**Keywords:** food choices, cooking intervention, food agency, diet quality, college students, healthy eating

## Abstract

Many college students struggle to cook frequently, which has implications for their diet quality and health. Students’ ability to plan, procure, and prepare food (food agency) may be an important target for shifting the college student diet away from instant and inexpensive staples like packaged ramen. The randomized intervention study included two sequential cooking interventions: (1) six weeks of cooking classes based in food agency pedagogy held once per week, and (2) six weekly home delivered meal kits (3 meals per kit) to improve food agency, diet quality, and at home cooking frequency of college students. Based on availability and subsequent randomization, participants were assigned to one of four conditions that included active cooking classes, meal kit provision, or no intervention. Participants who took part in the cooking intervention had significant improvement in food agency immediately following the intervention period. Participants who did not participate in cooking classes and only received meal kits experienced significant, though less pronounced, improvement in food agency scores following the meal kit provision. Neither intervention improved diet quality or routinely improved cooking frequency. Active cooking classes may improve food agency of college students, though further research is needed to determine how this may translate into improved diet quality and increased cooking frequency.

## 1. Introduction

Young adulthood is a critical time for developing independent living behaviors like cooking. Despite the importance of being able to cook, Larson et al. [1] found that the majority of young adults surveyed did not perform food preparation and purchasing behaviors such as buying fresh vegetables, writing a grocery list, or preparing dinner. In addition to being an important life skill, cooking may impact one’s health. Laska et al. found that home food preparation and meal regularity were the factors most associated with healthy dietary patterns in a sample of college students [2], and Larson et al. found that college students who reported that they frequently prepared food at home were more likely to meet fat, calcium, fruit, vegetable, and whole grain dietary goals [1]. Compared to young adults living on a college campus, young adults who live off campus report less healthy dietary intake and less healthy home food availability [2,3]. Previous research points to the importance of learning cooking skills at an early age. In a study of over 1000 adults, Lavelle et al. found that those who reported learning cooking skills as children or teens had more confidence in their cooking practices, attitudes, and better diet quality as adults [4]. However, additional research by Lavelle et al. [5] suggests that cooking skills are not often transferred from parents to children anymore, suggesting that deliberate cooking instruction may be necessary for children and young adults to learn how to cook. Developing cooking skills is additionally important when considering that results from a 10-year longitudinal study by Laska et al. found that engagement in food preparation during emerging adulthood was associated with healthier dietary intake during the mid–late twenties suggesting that exciting college students to cook may have long-term health benefits [6].

Issues such as lack of cooking knowledge, restricted access to healthy foods, economic constraints, and time scarcity have all been cited as potential factors for the decline in home cooking frequency seen in America since the 1960s [7,8,9,10]. Time scarcity seems to be an especially relevant factor for college students with several studies finding that college students identified time scarcity as a significant barrier to cooking food at home [1,11]. Although much work has illustrated that many people feel they cannot engage in health behaviors such as cooking because of perceived or actual time scarcity [8,12], very few intervention studies have examined potential strategies that might decrease people’s perceptions of time scarcity and increase the number of days they are willing to cook meals at home.

Cooking classes may be an advantageous way to address many barriers to cooking. Cooking classes can impact time scarcity by helping participants learn how to plan meals and easily prepare ingredients [7]. They may also teach budgeting skills to address economic barriers. Several studies on college students have examined course-based interventions where students in nutrition courses were either asked to cook a healthy entrée or a whole grain and then report back to the instructor or class on the experience [13,14]. Although students favorably reviewed each cooking activity, these interventions did not include any measures evaluating increase in cooking skills or dietary changes, and did not teach students how to cook. In another study, Levy and Auld found that college students who attended four cooking classes, had greater shifts in their attitudes and behaviors than students who attended four cooking demonstrations [15]. A review by Reicks et al. [16] on cooking and food preparation interventions for adults found that cooking interventions had a positive impact on main outcome measures such as diet quality and cooking confidence, but that more research using validated assessment tools was needed to truly assess the efficacy of cooking interventions.

Cooking classes may be an important intervention target because they can increase one’s food agency. Food agency is a measure of one’s ability to adapt in their cooking practice and overcome challenges such as limited time and money, imperfect physical environment, and even sensory challenges. Therefore, food agency examines cooking as more than just a manual skill. People with high food agency have the manual cooking skills to be successful, but they also have self-efficacy around planning and preparing to cook, as well as adapting during the cooking process [17]. Trubek et al. [17,18] designed a pedagogy focused on increasing food agency by incorporating instruction on the cognitive, technical, and mechanical skills necessary to successfully cook. Therefore, in this pedagogy, one would learn not only how to sauté a vegetable, but also the organizational and decision-making skills that would be useful no matter the physical environment or resources available while cooking. Using pedagogy designed to increase one’s food agency, the participant is better prepared to overcome potential daily challenges that could prevent them from meeting their cooking, nutrition, and social goals, they are more adaptable [18]. The food agency pedagogy then combines several behavior change techniques; providing information, providing instruction, and prompting practice that each individually commonly appear in cooking and food skills interventions with varying levels of efficacy [19].

Along with teaching cooking competency through a pedagogy that emphasizes food agency, it may be helpful to provide a material incentive to help overcome remaining barriers to healthy cooking for young adults. Incentives have been deployed successfully for a variety of health behaviors and health outcomes such as exercise and weight loss [20,21]. As provisioning food, deciding what to buy, and having the resources to buy it, are often cited as barriers to cooking at home, it may be the case that providing people with ingredients and/or recipes each week could serve as incentives to overcome the provisioning barrier and encourage people to implement the skills learned during their cooking classes. Two studies focused on families provided meal plans, recipes, and ingredients to cook meals at home and found that families reported preparing most of the meals, eating healthier, and feeling better, suggesting that material provision can positively impact cooking and nutrition [22,23]. Recently, meal kit delivery services such as Hello Fresh, Blue Apron, and Purple Carrot have risen in popularity [24]. These services provide subscribers with the raw ingredients in the correct amounts necessary to make particular recipes and then also provide the recipes. There is very little research exploring whether meal kits have a positive impact on cooking behavior or diet quality in any population. Two recent studies from Australia looked at the nutrition quality of meal kits. One found that meal kits contained three vegetables per meal which could help improve diet quality, but also more than the recommended amount of sodium per meal which could negatively affect diet quality [25]. The second found that meal kits were high in fat and sodium, but did provide adequate servings of vegetables [26].

The current study aimed to assess the efficacy of six weekly cooking classes followed by six weeks of meal kit delivery for improving food agency, diet quality, and frequency of cooking at home for college students living off campus. We assumed that students participating in the cooking classes would have greater improvements in food agency, diet quality, and frequency of cooking at home than students in the control group. Furthermore, we believed that meal kits would help sustain any gains in food agency, diet quality, and cooking frequency for those who had taken the cooking classes, and lead to greater gains in food agency, diet quality, and cooking frequency for those who received the meal kits than the control group.

## 2. Materials and Methods

### 2.1. Participants

The study was a 12-week randomized control trial with three treatment groups and a control group. Undergraduate students living off-campus with access to a kitchen were recruited to participate. Inclusion criteria included being an actively enrolled student at the University of Vermont and aged 18–25. Exclusion criteria included living with parents or guardians and a self-report of cooking an average of more than 3 dinners at home per week, as we wanted to recruit students who did not already cook frequently. Recruitment was conducted from September 2019 to October 2019 and was approved by the Committee on Human Research in the behavioral Sciences at the University of Vermont. The study was registered with ClinicalTrials.gov, registration number: NCT04084028.

### 2.2. Screening Procedures

The study contained four intervention groups, which were defined as cooking intervention followed by meal kit intervention (Cook + MK); cooking intervention with no meal kit intervention (Cook Only); no cooking intervention followed by meal kit intervention (MK Only); and Control, no cooking or meal kit interventions. Those in the control condition received no cooking support over the course of the study. They served as a benchmark for how college students normally cook without focused instruction.

Convenience sampling was used to recruit study participants. Interested students completed an online application and if deemed eligible, were invited to an in-person information session where informed consent was obtained. Enrolled participants were assigned to one of four scheduled class periods based on their stated availability. Each class included 8–9 students, sample size was limited by the capacity of the teaching kitchen used in the study. A random number generator was used to determine the group assignment (Cook + MK or Cook Only) for each class period. Study applicants who were not available for the scheduled cooking classes were invited to participate in the groups not receiving cooking classes. Students who chose this option were assigned to either the MK Only or Control group by a random number randomization scheme. Following all randomization procedures, participants were assigned as follows: Cook + MK, *n* = 18; Cook Only, *n* = 16; MK Only, *n* = 9; Control, *n* = 10.

### 2.3. Cooking Intervention

Six cooking classes were held every week for six consecutive weeks during the fall semester of 2019 for the Cook + MK and Cook Only groups. This six-week period of cooking classes reflected phase 1 of the study. The cooking classes were patterned after Dr. Amy Trubek’s pedagogy that teaches culinary skill using the food agency philosophy. This is known as the food agency pedagogy, which has emerged from long-term interdisciplinary research [7,17,18,27]. Classes included a brief lecture on the day’s topic, a laboratory session in which participants worked in teams of two to actively practice skills and cook a meal, and ended with time to taste the meal and conduct a sensory analysis. Classes were taught by a chef educator trained in the food agency pedagogy. In line with the food agency pedagogy, each class focused on developing three “understandings” for participants. The first focused on the importance of sensory analysis for helping one engage and connect with the cooking process and broaden their knowledge of various ingredients and cooking methods. The second understanding focused on the importance of knife skills and how knife skills are integral to creating a cooked meal from raw ingredients. Finally, the importance of mise en place, or organization of space, ingredients, thought, tools, and processes, was emphasized in each class. In addition to the three understandings that were emphasized in each class, class topics included: exploring whole grains; herbs and spices; vegetables—raw, blanched, or roasted; cooking animal protein; fats and flavor; and baking basics. Class recipes such as coconut noodle bowl and Indian curry chickpeas with cucumber raita were designed to be relevant for an audience of college-aged participants while simultaneously highlighting basic food agency skills.

### 2.4. Meal Kit Intervention

Following the 2019/2020 winter break, approximately eight weeks after the conclusion of Phase 1 cooking classes, meal kit boxes were delivered directly to the homes of participants in the Cook + MK and MK Only groups, initiating the beginning of Phase 2 of the study. Participants received meal kits for six consecutive weeks. Each box contained all of the ingredients and instructions needed to prepare three meals designed to feed two people. Each week, participants were able to select meals that suited their dietary needs (vegetarian, vegan, gluten-free) from a list of eighteen options.

### 2.5. Dependent Measures

Data were collected via emailed survey at baseline and following each of the two study intervention periods. Participants were offered Amazon gift cards, gift cards to local stores, and kitchen tools (culinary knives, cookbooks, cookware, baking equipment, etc.) to complete surveys at each time point. At each data collection time point, participants were emailed a link to a set of online surveys that included all dependent measures.

Cooking and Food Provisioning Action Scale (CAFPAS). The CAFPAS scale is designed to measure food agency including one’s ability to set and meet cooking goals. The CAFPAS is composed of 28 items corresponding to three subscales: Food Self-Efficacy, Food Attitude, and Structure. Although a relatively new measure, the CAFPAS scale has reported adequate internal validity and test-retest reliability [27].

The Cooking Perceptions and Behaviors questionnaire. This questionnaire is a 53-item survey measuring three factors: Perceptions of Cooking, Cooking Confidence and Attitudes, and Cooking Behaviors. This survey was primarily used in the current study to assess the average frequency each week that participants cooked breakfast, lunch, and dinner at home [28].

Dietary Quality. Dietary recalls were recorded using the Automated Self-Administered 24-Hour Dietary Assessment Tool (ASA24). Participants were asked to complete three 24-h recalls at each time point, which included two weekdays and one weekend day. The ASA24 is a web-based tool that walks users through a multi-pass dietary recall. The ASA24 data were then used to calculate Healthy Eating Index (HEI) scores for each participant at each time point. The HEI is a measure of overall diet quality. The HEI can assess one’s compliance with the U.S. Dietary Guidelines as well as changes in dietary patterns over time [29].

### 2.6. Statistical Analysis

Repeated measures analysis (SAS, PROC MIXED) was used to assess changes over time in Cooking and Food Provisioning Action Scales, Healthy Eating Index scores, and the average number of meals cooked at home. Linear contrasts were constructed to assess within-group changes during phases 1 and 2 and to compare treatment groups on the magnitude of change. All analyses were performed using SAS Version 9 statistical software (SAS Institute, Cary, NC, USA). Statistical significance was based on α = 0.05.

## 3. Results

### 3.1. Participants

Fifty-three university students were enrolled as study participants and assigned to one of four intervention conditions (See Figure 1). Participants were primarily white, non-Hispanic, female, and juniors or seniors in college (See Table 1). Participants in both the Cook + MK and Cook Only groups attended cooking classes regularly. Attendance for each of the four classes ranged from 83% to 93% over the six-week period.

### 3.2. Cooking and Food Provisioning Action Scales (CAFPAS)

Participants in the Cook Only and Cook + MK conditions had significant increases in their total CAFPAS scores during the cooking intervention phase of the study (See Table 2). The combined Cook Only and Cook + MK groups increase in CAFPAS score during phase 1 was significantly greater than the increase in the combined MK Only and Control groups. During phase 2, there were no significant differences in the magnitude of change in CAFPAS scores between the two cooking groups or between the Control and MK Only groups, although the MK Only group had a significant increase in their CAFPAS total score during Phase 2.

### 3.3. Cooking Frequency

The only significant change in cooking frequency was for the Cook + MK Group. In phase 2 of the study, the Cook + MK group significantly increased the number of breakfasts, lunches, and dinners they cooked at home (See Table 2).

### 3.4. Diet Quality

There was a significant decrease in Healthy Eating Index (HEI) scores for the Cook + MK group during phase 1, but there was no significant HEI change for the Cook Only group during phase 1. Furthermore, there was no significant difference between HEI score change for the cooking groups versus the non-cooking groups in phase 1. There were no other significant changes in HEI scores within groups during either phase. Finally, there were no significant differences in the changes in HEI scores between the Cook + MK vs. Cook Only groups and the MK only vs. Control groups in phase 2 (See Table 3).

## 4. Discussion

Results indicate that providing cooking classes to college students living off campus on their own helped to increase their food agency. Both conditions who attended cooking classes saw significant increases in their CAFPAS scores compared to the conditions that did not receive cooking classes. During phase 2 of the study, food agency scores remained stable for the two conditions that had received cooking classes in phase 1. For the condition that received meal kits only, their CAFPAS scores increased during the meal kit intervention (phase 2), but not as much as the CAFPAS increased for the cooking class interventions in phase 1. From these results, it seems that cooking classes designed to increase food agency were more effective than providing meal kits at increasing food agency. These results would seem to validate that the food agency curriculum as developed by Trubek et al. [17] specifically targets food agency and does indeed help empower learners to plan, prepare, and execute meal preparation activities. The pedagogy encouraged the development of self-efficacy around cooking and the ability to better navigate material, physical, and cognitive barriers, all of which participants in the study and young adults in general may face as cooking novices. Although there have been several previous cooking class interventions in a college population [13,14,30], none have been focused on increasing one’s self-efficacy around cooking and overcoming barriers to planning, preparation, and execution that have been noted in young adults [1,10]. This is one of only two studies [31] that have implemented the food agency pedagogy and then evaluated change in food agency, so it is encouraging that the pedagogy is effective.

It is interesting that although not specifically designed to impact food agency, the provision of meal kits did have a positive impact for those who had not previously taken the cooking classes. Perhaps the ingredient and recipe provision involved in meal kits helped participants bolster their feelings of self-efficacy around cooking as well. There has been very little research surrounding meal kits and how they may influence one’s cooking skills; to our knowledge, this is the first study to examine the impact of providing meal kits on food agency. However, meal kits are designed to decrease the need for advance planning, as well as increase one’s cooking knowledge, decrease the time devoted to cooking, and make the cooking experience more enjoyable overall [32]. By removing the need for as much planning and preparation while also helping participants avoid “kitchen disasters” by providing simple, step-by-step instructions, meal kits may help novice cooks build self-efficacy around cooking and increase their food agency slightly, even when the meal kits may not formally help develop the “three understandings” of food agency focused around sensory experience, knife skills, and mise en place [17]. In a meal kit intervention with families, Horning et al. [33] found that meal kit provision led to increases in cooking self-efficacy and healthy food availability, although there was no change in fruit and vegetable intake. The increase in cooking self-efficacy is similar to the increase in food agency observed with meal kits in the current study, and the lack of impact on dietary intake is also similar between the studies.

The mean baseline CAFPAS scores for all conditions were between 11.64 and 12.66; these are substantially lower than the average score of 14.1 for people in their twenties reported in previous research [27], but similar to baseline scores, 11.25, of older adults who were infrequent home cooks in a previous study our group conducted [31], as well as the average score of 12.94 for adults with some college education recently reported in another study [34]. Therefore, it does seem that participants in our study had low food agency at baseline, verifying perhaps that we had recruited the students we were aiming to recruit, those who did not cook frequently and did not feel confident cooking. CAFPAS scores for the cooking class groups improved just over 2 points to around 14 (see Table 2), while scores in the meal kit only group improved 1.6 points to just over 13. These scores are more in-line with the scores found in previous research for people in their 20s [27].

Unfortunately, increases in food agency scores over phase 1 did not translate to increased cooking frequency for breakfast, lunch, or dinner over phase 1 for either of the cooking conditions. Therefore, it seems that even though food agency improved, it was not directly related to subsequent increases in how often students cooked at home. Perhaps even though students felt more confident about their cooking skills, they still had to navigate barriers such as time constraints, budgetary constraints, or material/space constraints that prevented them from cooking more frequently at home. Previous research certainly affirms that college students struggle to cook because of limited time, money, and materials as well as cooking knowledge [1,11]. Although our cooking classes were designed to address knowledge and skills, and we were hoping that increased cooking skills would translate into less time spent cooking and greater ability to cook with limited financial and material resources, it may be that the food agency pedagogy needs an even clearer focus on time savings and low-budget cooking to really translate increased food agency into more frequent cooking practice. However, previous research has shown that higher food agency scores are related to greater cooking frequency [27,31,34]. Another possible explanation is that although food agency increased, it was still not high enough to translate into ease of cooking at home and therefore more frequency of cooking from home. Perhaps additional cooking instruction and guided practice would lead to further increases in food agency and subsequent translation into increased “real-world” cooking.

Only the Cook plus Meal Kit condition showed any significant changes in frequency of cooking at home. During phase 2, the Cook plus Meal Kit condition had significant increases in the number of breakfasts, lunches, and dinners cooked at home. The provision of meal kits may have been responsible for some of this increase, especially around lunches and dinners, but it is curious then that the Meal Kit Only group did not also have significant increases in cooking frequency during phase 2. Perhaps participants in the Cook plus Meal Kit group were able to better use the meal kits because of the knowledge gained during their time in cooking classes and therefore had a greater frequency of cooking, but with these disparate results in the two meal kit groups it is difficult to conclude that meal kits had a positive impact on cooking frequency.

Finally, increased food agency scores did not translate into increased Healthy Eating Index (HEI) scores. There was only one significant change in HEI scores and it was a decrease during phase 1 for the Cook plus Meal Kit condition. In fact, during both phases of the study, all changes in HEI scores for all groups were decreases. This pattern may reflect the flow of a college semester. End of phase 1 data were collected right before final exams which may be a time where students are very stressed and diet quality suffers. Phase 2 data were collected right before spring break which again may be a time where student stress is high and diet quality decreases. Although we can only speculate on the causes of decreased diet quality scores, it does not seem that increased food agency led to increased diet quality for college students, whereas a previous study by Wolfson et al. [34] did find associations between higher food agency and more frequent consumption of fruits and vegetables in an adult sample. While all recipes used in the cooking intervention reflected principles of a nutritious diet, and the chef educator spoke about nutrition and balanced diet at each class, perhaps adding specific language to the food agency pedagogy addressing basic nutrition principles like including fruits, vegetables, legumes, and whole grains in your dietary pattern might be a strategy for increasing both food agency and dietary quality. It may also be that since increased food agency scores did not translate to increased cooking frequency, students were not implementing the nutrition principles they learned in the food agency pedagogy in their daily lives. Recent research on the nutrient composition of meal kits also indicates that their nutrient composition may not always align with dietary guidelines, and therefore they may not lead to increased diet quality depending on the options selected [25,26]. Finally, since the study took place during the winter months in the northeastern United States, it is possible that students were choosing to eat foods and meals that were more hearty and comforting, and that if the study had taken place over the summer, students would have been eating more seasonally available produce and generally lighter meals that would have resulted in higher HEI scores. Recent research in older women found that although diet quality was lowest in the winter, differences in HEI scores between seasons were not substantial enough to be considered in studies of diet quality [35], indicating that seasonality was probably not a large driver to the diet quality results observed in this study.

The results of the current study suggest that a cooking intervention focused on food agency does in fact increase food agency; however, future research may be needed to better understand how increases in food agency could then translate to both increased frequency of cooking from home for young adults as well as increased diet quality. Future research could add a more deliberate nutrition education component to the food agency pedagogy, or perhaps focus more explicitly on ways to overcome barriers college students face when trying to increase their cooking practice like time management and limited resources. Perhaps in addition to raising self-efficacy around cooking, helping students organize a weekly cooking plan based on what tools, time, and food they have available would help increase cooking frequency. The fact that the Cook + Meal Kit group did see some increases in cooking frequency suggests that pairing the food agency pedagogy with material support may be impactful for increasing cooking frequency. Measuring diet quality is notoriously difficult, and future research could also focus on other ways to capture changes in diet related to cooking interventions, as well as any mental/physical health benefits that developing a cooking practice may impart. Finally, more research is needed on the impacts of providing meal kits both on cooking practice and diet quality. Future research could examine who benefits most from meal kit provision, whether certain types of meal kits are most effective in building cooking practice or increasing diet quality, and consumers’ perceptions of meal kits.

### Strengths and Limitations

This study has several important strengths, including the novelty of employing both the food agency pedagogy and meal kit interventions in a young adult population. The use of the food agency pedagogy and a trained chef educator to implement it helped ensure that participants in the cooking groups all received a standard curriculum designed to directly impact their food agency. Additional strengths include using validated tools such as the CAFPAS, ASA24, and HEI to measure food agency and diet quality. The sample size of the study may have limited the power to detect small differences in the outcome measures, future studies could focus on increasing sample size. The current study was restricted by space in the kitchen lab, as well as student schedules that made participant availability for cooking classes more limited. Because of these constraints, we had to modify our randomization schema which may have impacted results, although we did not find baseline differences in outcome measures across groups. We also had several participants who did not complete data collection throughout all phases of the study; this may have influenced our ability to accurately determine differences in outcome measures. Additionally, because the study only examined two six-week phases of time, we are not able to determine whether any increases in food agency persist across time. Unfortunately, we do not know the nutrition information of the meals that participants selected during the meal kit intervention. Because there were 18 meal choices each week, participants were choosing from a large variety of meals, which may have had various impacts on diet quality. Future studies could more closely examine the nutrition content of meals selected during a meal kit intervention. Finally, the rhythms of the college calendar may have impacted our results and results may have been different with a young adult population that was not on the schedule of an academic semester.

## 5. Conclusions

The findings of this study indicate that a cooking class intervention focused on increasing food agency for young adults may have a positive impact on food agency, but increases in food agency did not translate into increases in dietary quality or the frequency of cooking at home. Therefore, future studies are needed to determine how improvements in food agency may lead to improvements in diet quality or greater cooking frequency at home for young adults. Furthermore, although meal kits are often marketed as a way to help consumers cook more frequently at home, our study did not find that meal kits had consistent positive impacts on cooking frequency or diet quality. Further examination of how meal kits may impact cooking behavior and food agency is needed. Although our results did not provide evidence that feeling more confident in one’s cooking ability translated into more frequent cooking or increased diet quality, building self-efficacy around cooking is still an important outcome for young adults who will be able to utilize their cooking skills throughout their lives.

## Figures and Tables

**Figure 1 nutrients-13-01674-f001:**
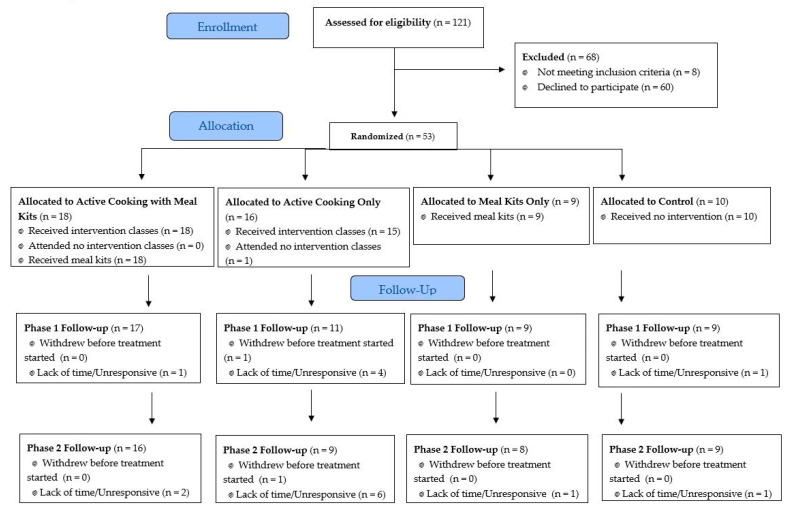
Consort flow diagram.

**Table 1 nutrients-13-01674-t001:** Demographic characteristics of participants.

	Cook + MK (*n* = 18)	Cook Only (*n* = 16)	MK Only (*n* = 9)	Control (*n* = 10)
Age, years (mean ± SD)	20.6 ± 1.3	20.9 ± 1.2	20.6 ± 0.5	20.6 ± 0.8
University standing				
Junior	8 (44%)	4 (25%)	2 (22%)	4 (40%)
Senior	7 (39%)	12 (75%)	7 (78%)	6 (60%)
Other	3 (17%)	0	0	0
Gender (*n*; %)				
Female	12 (67%)	13 (81%)	7 (78%)	9 (90%)
Male	4 (22%)	3 (19%)	2 (22%)	1 (10%)
Non-binary/ third gender	1 (6%)	0	0	0
Prefer not to answer	1 (6%)	0	0	0
Race (*n*, %)				
Non-Hispanic White (*n*; %)	14 (78%)	14 (88%)	9 (100%)	10 (100%)
East Asian or Asian American	2 (11%)	0	0	0
Latino or Hispanic American	1 (6%)	0	0	0
Black, Afro-Caribbean, or African American	0	1 (6%)	0	0
South Asian or Indian American	1 (6%)	0	0	0
Bi-racial	0	1 (6%)	0	0
Relationship status (n, %)				
Single	18 (100%)	14 (88%)	8 (89%)	10 (100%)
Living with partner	0	2 (12%)	1 (11%)	0
Employment status				
10–20 h per week	5 (28%)	3 (19%)	4 (44%)	2 (20%)
<10 h per week	7 (39%)	5 (31%)	4 (44%)	7 (70%)
Not employed	6 (33%)	8 (50%)	1 (11%)	1 (10%)

Note. Percentages have been rounded to nearest whole percent and may not add up to 100%. MK, meal kit; SD, standard deviation.

**Table 2 nutrients-13-01674-t002:** Mean food agency (CAFPAS) scores and mean meals cooked per week at baseline and change following each intervention phase.

Group	Time	CAFPAS Score (LSMean ± SE)	Breakfasts (LSMean ± SE)	Lunches (LSMean ± SE)	Dinners (LSMean ± SE)
Cook + MK (*n* = 18)	Baseline	11.70 ± 0.47	1.78 ± 0.43	1.33 ± 0.40	2.70 ± 0.40
	Phase 1 Change	2.03 ^1^ ± 0.37	0.778 ± 0.50	0.333 ± 0.40	0.778 ± 0.45
	Phase 2 Change	−0.40 ± 0.38	1.42 ^2^ ± 0.52	0.853 ^3^ ± 0.41	1.45 ^4^ ± 0.47
Cook Only (*n* = 16)	Baseline	11.85 ± 0.50	1.75 ± 0.46	0.750 ± 0.42	3.38 ± 0.42
	Phase 1 Change	2.03 ^1^ ± 0.46	0.313 ± 0.61	0.898 ± 0.48	0.232 ± 0.54
	Phase 2 Change	−0.21 ± 0.59	−0.072 ± 7.9	0.882 ± 0.62	−0.459 ± 0.71
Cook + MK vs. Cook Only (*n* = 34)	Phase 2 Change Comparison	−0.20 ± 0.70			
MK Only (*n* = 9)	Baseline	11.64 ± 0.67	2.67 ± 0.61	2.44 ± 0.56	4.11 ± 0.56
	Phase 1 Change	0.007 ± 0.54	0.44 ± 7.1	−0.222 ± 0.55	0.111 ± 0.63
	Phase 2 Change	1.60 ^4^ ± 0.55	0.115 ± 0.74	0.960 ± 0.57	0.108 ± 0.66
Control (*n* = 10)	Baseline	12.66 ± 0.63	4.40 ± 0.58	2.90 ± 0.53	4.70 ± 0.53
	Phase 1 Change	0.87 ± 0.51	−0.307 ± 0.70	0.558 ± 0.54	0.839 ± 0.62
	Phase 2 Change	0.68 ± 0.52	0.333 ± 0.71	0.444 ± 0.55	0.111 ± 0.63
MK Only vs. Control (*n* = 19)	Phase 2 Change Comparison	0.92 ± 0.75			
(Cook + MK and Cook Only) vs. (MK Only and Control) (*n* = 53)	Phase 1 Change Comparison	3.18 ± 0.95			

^1^*p* < 0.0001, ^2^
*p* = 0.008, ^3^
*p* = 0.038, ^4^
*p* = 0.003. CAFPAS, Cooking and Food Provisioning Action Scale; LSMean, least square mean; SE, standard error.

**Table 3 nutrients-13-01674-t003:** Mean baseline Healthy Eating Index (HEI) scores and score changes at phase 1 and phase 2.

Study Group	Baseline Score (LSMean + SE)	Phase 1 Change (LSMean + SE)	Phase 2 Change (LSMean + SE)
Cook + MK (*n* = 18)	65.11 ± 3.17	−7.45 ^1^ ± 3.09	−4.65 ± 3.37
Cook Only (*n* = 16)	61.84 ± 3.37	−3.52 ± 3.80	−2.07 ± 4.42
MK Only (*n* = 9)	61.08 ± 4.49	−7.26 ± 4.37	−5.70 ± 4.55
Control (*n* = 10)	67.84 ± 4.49	−4.74 ± 4.57	−4.56 ± 4.63
Cook + MK vs. Cook Only (*n* = 34)		n/a	−2.58 ± 5.56
Mk Only vs. Control (*n* = 19)		n/a	−1.14 ± 6.49
(Cook + MK and Cook Only) vs. (MK Only and Control) (*n* = 53)		1.03 ± 8.00	n/a

^1^ *p* = 0.01. LSMean, least square mean; SE, standard error.

## Data Availability

Not applicable.

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
