# Peer review of "Beyond Ramen: Investigating Methods to Improve Food Agency among College Students"

_nutrients, 2021, doi:10.3390/nu13051674_

Round 1

Reviewer 1 Report

This is an interesting study which investigates methods to improve cooking (and hopefully healthy eating) in college students who do not usually cook frequently. The number of participants is rather small, which is normal for this kind of study. Unfortunately the drop out rate of participants was rather big, which made making conclusions by the authors challenging. Nevertheless, this is an interesting and important topic.

Comments:

1) Could you please better explain Phase 1 and Phase 2 in the Method section, since this is not very clear.

2) The first paragraph of the chapter 2.2. Screening Procedures repeats in the third paragraph.

3) When was the Meal Kit Intervention done? How long after the cooking classes?

4) Please check in the Figure 1. the numbers of participants included in each phase. Some numbers are missing and some don't match.

Minor revision

1) line 14,15 - aren't meal kit provision and cooking classes interventions? this is confusing.

2) line 95 and 97 - change "believed" in "assumed"

Reviewer 2 Report

The present paper is interestingly approaching an actual public health issue, nutrition and cooking skills and more than that, the so called food agency, id est the (off-campus student’s) ability to plan, prepare and consume healthy home cooked meals in order to improve the healthy eating for a better nutrition. - an important theme of the public health and nutrition.

The study was interventional, with (1) a pedagogy intervention (cooking classes) and (2) provision of the meal kits for  in four groups. The endpoints were evaluated using (new) standardized questionnaires (Healthy Eating Index), Coking and Food Provisioning Action Scale (CAFPAS), Dietary Recalls,  .

  1. Title

Beyond Ramen: Investigating Methods to Improve Food Agency among College Students”

 Title uses two terms that are not very familiar,: “food agency” and “ramen” – the terms would be recommended to be briefly explained in the abstract

  1. Keywords: food choices; cooking intervention; food agency; diet quality; college students

– may also contain healthy eating

  1. Abstract
  • Line 11 - Title uses two terms that are not very familiar,: “food agency” and “ramen” –it would be recommended the terms to be explained in the abstract (examples “Food agency” is one’s ability to procure and prepare food within the contexts of one’s social, physical, and economic environment” [Wolfson JA, Lahne J, Raj M, Insolera N, Lavelle F, Dean M. Food Agency in the United States: Associations with Cooking Behavior and Dietary Intake. Nutrients. 2020 Mar 24;12(3):877. doi: 10.3390/nu12030877].
  • The specific university joke about the packs of ramen (a Japanese form of pasta) of the poor university students should be briefly addressed, otherwise the internationally reader may be a little confused (14 J. Food L. & Pol'y 154 (2018) The End of the Ramen Diet: Higher Education Students and SNAP ). Benefits_.
  • Line 13 – a short description of the study groups (n) and intervention would be welcomed (e.g. we have performed an interventional randomized study on college students living off-campus (n=)
  • Line 15 – study’s objectives may be better specified: pedagogic intervention and meal kits provision in order to improve the (a) food agency scores (b) diet quality (c) at home cooking frequency in college students
  • Line 17 – a more concrete description of the intervention would be recommended (e.g. meals were provided every week – quantity for one week, after each (weekly) cooking class, during a six weeks period.)
  1. References
  • Most of reference actual, however, actual research references (not older than ten years- as in 20% of actual references) would be welcomed;
  • Six of 24 citations contain names of authors, that may be eventually explained by the originality of the subject approached
  1. Introduction
  • text need minor editing services
  • line 66 - this phrase may be suited also to be briefly exposed in the abstract
  • Knowledge gap mentioned
  • Objective of the study – may be added “focused on off campus students with access to a kitchen ‘
  1. Materials and methods
  • Line 131, 135 – what random assignment table was used?
  • Methodology description - missing:
    • sample size calculation,
    • sample selection method,
    • line 109 - representativeness of the sample/ statistical representability of the target group
  • Questionnaire
  • Questionnaire may be added - as supplementary data/annex
  1. Results
  • Line 200 – white non-hyspanic was an inclusion criteria?
  • Line 240 – living off campus without guardians
  1. Please add if any:
    1. Limitations subsesction - (line 339)

Reviewer 3 Report

It is well written and timely topic. Considering not much impact in this study, it is better to link with health impact. In conclusions, it can be stated to link with health, may be with their BMI, weight reduction efforts, this will be a useful tool in the future. It may be useful to include something related to sustainability in conclusion section. 

Author Response

Thank you for your thoughts on our manuscript. Although it may be interesting to link our study intervention, measures, or outcomes with weight or health, we do not feel we can make conclusions regarding the impact on weight or a general concept of health because we did not have measures addressing these outcomes.  Our diet quality measure was meant to measure one form of health, eating behavior, and we did not see significant benefits of the intervention on this outcome measure.  We agree that future research should further explore connections between cooking behavior and health outcomes.  We do discuss the lack of sustained behavior change in our discussion section.